# Allopregnanolone Promotes Migration and Invasion of Human Glioblastoma Cells through the Protein Tyrosine Kinase c-Src Activation

**DOI:** 10.3390/ijms23094996

**Published:** 2022-04-30

**Authors:** Carmen J. Zamora-Sánchez, Claudia Bello-Alvarez, Mauricio Rodríguez-Dorantes, Ignacio Camacho-Arroyo

**Affiliations:** 1Unidad de Investigación en Reproducción Humana, Instituto Nacional de Perinatología-Facultad de Química, Universidad Nacional Autónoma de México, Cuidad de México 04510, Mexico; carmenjzamora@gmail.com (C.J.Z.-S.); clautus.bello@gmail.com (C.B.-A.); 2Laboratorio de Oncogenómica, Instituto Nacional de Medicina Genómica, Ciudad de México 14610, Mexico; mrodriguez@inmegen.gob.mx

**Keywords:** allopregnanolone, progesterone, neurosteroid, steroidogenesis, Aldo-keto reductase, c-Src, glioblastoma, glioma, cancer progression

## Abstract

Glioblastomas (GBs) are the most aggressive and common primary malignant brain tumors. Steroid hormone progesterone (P4) and its neuroactive metabolites, such as allopregnanolone (3α-THP) are synthesized by neural, glial, and malignant GB cells. P4 promotes cellular proliferation, migration, and invasion of human GB cells at physiological concentrations. It has been reported that 3α-THP promotes GB cell proliferation. Here we investigated the effects of 3α-THP on GB cell migration and invasion, the participation of the enzymes involved in its metabolism (AKR1C1-4), and the role of the c-Src kinase in 3α-THP effects in GBs. 3α-THP 100 nM promoted migration and invasion of U251, U87, and LN229 human-derived GB cell lines. We observed that U251, LN229, and T98G cell lines exhibited a higher protein content of AKR1C1-4 than normal human astrocytes. AKR1C1-4 silencing did not modify 3α-THP effects on migration and invasion. 3α-THP activated c-Src protein at 10 min (U251 cells) and 15 min (U87 and LN229 cells). Interestingly, the pharmacological inhibition of c-Src decreases the promoting effects of 3α-THP on cell migration and invasion. Together, these data indicate that 3α-THP promotes GB migration and invasion through c-Src activation.

## 1. Introduction

Glioblastomas (GBs) are the most common tumors in the central nervous system (CNS) malignancies and have the worst overall survival, which makes them clinically relevant. GBs are often diagnosed at 50–65 years old [1]. According to data between 1995 and 2021, they have an incidence of 3–7 per 100,000 inhabitants depending on the nation, even though their frequency rises in many countries [2,3,4]. The survival time of patients with GBs is about 2–4 months without treatment [4,5] due to the highly proliferative and invasive capacity of GB cells [1]. GBs present a higher incidence in men than in women at 1.60:1, which remain unchanged for years [3,4]. This datum has brought attention to the study of sex-specific components that contribute to GB development, such as steroid hormones [6]. Gonadal steroids, like progesterone (P4), participate in the progression of GBs. At physiological concentrations, P4 promotes proliferation, migration, and invasion of GB cells, and it also augments the number of primary gliomaspheres [7,8,9,10,11]. In addition, it has been reported that GB cells actively metabolize P4 [12,13,14], although the effects of its reduced metabolites have been scarcely studied in cancer. Regarding the effects of 5α-reduced P4 metabolites, only a few studies on breast and ovarian cancers have been made [15,16,17].

The first step in the synthesis of active P4 metabolites is the irreversible reduction of P4 to 5α-dihydroprogesterone (5α-DHP) through the enzyme 5α-reductase (5αR) [12,13,18]. Then, 5α-DHP is reversibly reduced to allopregnanolone (3α-THP) by the 3α-hydroxysteroid dehydrogenases (3α-HSD), whose activity has been reported in four members of the Aldo-keto reductase enzyme superfamily (AKR1C1-4 by their coding genes) [19,20].

The steroid 3α-THP has been, by far, the most studied neuroactive metabolite of P4. 3α-THP is synthesized in the CNS by glial progenitors, neurons, and glial cells in different brain regions such as the cortex, olfactory bulb, hippocampus, thalamus, cerebellum, and amygdala [13,19,21,22,23]. 3α-THP increases the expression of proliferating markers, along with the cellular proliferation of rat and human neural progenitor cells [24]. Besides, it has cytoprotective properties against cytotoxic insults in oligodendrocytes [25]. Interestingly, it promotes migration and changes in the morphology of rat Schwann cells through the activation of kinases like c-Src and FAK [26]. 3α-THP also promotes proliferation and the expression of different cell cycle and cytoskeleton regulators, such as the Rho-associated protein kinase (ROCK) in the U87 human GB cell line [18,27].

P4 metabolism enzymes and the active production of 3α-THP have been reported in GB cells [12,14,18,28]. In human-derived GB cell lines, it has been reported that P4, 5α-DHP, and 3α-THP promote GB malignancy by increasing cell proliferation. Moreover, this effect could be independent of the classical P4 receptor (PR), particularly in the case of 3α-THP, since it has a very low affinity to PR [18,27,28].

The proto-oncogene non-receptor tyrosine kinase c-Src is a hub protein involved in many cell transduction signals, promoting inflammation, cell survival, proliferation, migration, invasion, and tumor resistance to treatment in GB [29,30,31]. As we have mentioned, some of the effects of 3α-THP are mediated by c-Src. However, the 3α-THP effect on GBs progression is poorly known. Here, we report the effects of 3α-THP on cell migration and invasion of different human-derived GB cell lines, the relevance of its metabolism, and the interplay between c-Src and 3α-THP in GB progression. Consult the abbreviations of this manuscript to see a list of the most used abbreviations in this study.

## 2. Results

### 2.1. 3α-THP Promotes Migration and Invasion of Human GB Cells

First, we determined the effect of different concentrations of 3α-THP (10 nM, 100 nM, and 1 µM) on cell migration of the three human GB cell lines U251, U87, and LN229. The effect of 3α-THP was compared to P4 10 nM, due to its previously reported promotion of migration in these cells [8]. In Figure 1, we show that P4 10 nM promoted the migration of U251 cells after 12 h of treatment. Also, we observed that all concentrations of 3α-THP promoted the migration in U251 and U87 cells at 24 h (Figure 1a–c and Appendix A). In LN229 cells, only 3α-THP 100 nM increased migration at 24 h of treatment (Figure 1d and Appendix A). In all cell lines, 3α-THP 100 nM promoted cell migration; therefore, we used this concentration in the subsequent experiments.

Then, we evaluated the invasion capacity of the different human GB cell lines treated with 3α-THP 100 nM for 24 h. The invasion assays showed that P4 10 nM significantly increased the number of invading cells in U251 and LN229 lines but not in U87 cells (Figure 2), whereas 3α-THP 100 nM significantly increased the number of invading cells in the three evaluated cell lines (Figure 2a–f). In U251 and LN229 cells, the effect of 3α-THP was similar to that of P4 (Figure 2a,b,e,f).

### 2.2. Human GB Cell Lines Express 3α-THP Metabolism Enzymes

3α-HSD are necessary for 3α-THP metabolism. Previously, we have reported, the expression of the Aldo-keto reductases at the mRNA level, called AKR1C1-4 by the name of their genes (35): AKR1C1 (20αHSD), AKR1C2 (3α-HSD type 3), AKR1C3 (3α-HSD type 2), AKR1C4 (3α-HSD type 1) [32,33]. They have a high amino-acid sequence identity, and a similar weight (32–46 kDa) [34]. Here, we determined the protein content of such enzymes in human GB cell lines and compared it with that of normal human astrocytes (HA). All of them were detected as a single Western blot band with a primary antibody designed for the 1–78 amino acids at their *N*-terminus. The content of AKR1C1-4 enzymes of the HA lysate and U87 cells was low and nearly detected (Figure 3), whereas their content was significantly higher in U251, LN229, and T98G cells when compared with HA (Figure 3).

### 2.3. Role of AKR1C1-3 Silencing in the Effect of 3α-THP on GB Cell Migration and Invasion

To determine the effect of 3α-THP metabolism in the migration of human GB cells in the U251 cell line, we silenced the expression of the Aldo-keto reductases AKR1C1-3, which mainly participates in metabolizing 3α-THP in the brain. First, we ensured that AKR1C1-3 isozymes were silenced during the wound healing assays (Figure 4a). 3α-THP promoted cell migration in the sAKR1C1-3 U251 cells and the Control siRNA group compared to the V sAKR1C1-3 and Control siRNA groups (Figure 4b,c); this indicates that 3α-THP promotes cell migration of U251 cell line independently of its metabolism by the enzymes AKR1C1-3.

Next, we determined the effect of 3α-THP metabolism on the human GB U251 cell line invasion. 3α-THP promoted the invasion of the U251 cell line in Control siRNA, and sAKR1C1-3 groups when compared with V (Control siRNA and sAKR1C1-3). We did not find any significant differences in cell invasion between groups treated with V (Control siRNA and sAKR1C1-3), nor between those treated with 3α-THP (Control siRNA and sAKR1C1-3) (Figure 5a,b). This points out that 3α-THP promotes cell invasion per se and not by generating other progestin products from AKR1C1-3 activity in GB cells.

### 2.4. 3α-THP Promotes c-Src Activation in Human GB Cell Lines

The involvement of c-Src in the transduction signals induced by 3α-THP has been reported [26]. To elucidate the participation of c-Src in the effects of 3α-THP on human GB cells, we first determined the phosphorylation of Y416 residue of c-Src at (5–60 min). Phosphorylation increased at Y416 of c-Src residue was correlated with an augment of c-Src activity. We found a significant augment in Y416 phosphorylated c-Src after 10 min of 3α-THP 100 nM treatment in U251 cells compared to V (Figure 6). We also observed a significant increase in Y416 phosphorylated c-Src with 3α-THP treatment at 15 min in U87 and LN299 cells (Appendix A).

### 2.5. Influence of c-Src Inhibition in the Effect of 3α-THP on Cell Invasion

Since we observed changes in c-Src phosphorylation, then we evaluated if 3α-THP effects on invasion were mediated by this enzyme. We performed invasion assays with the c-Src pharmacological inhibitor PP2 1 μM, alone or in combination (3α-THP + PP2). Again, we showed that 3α-THP promoted invasion in U251 and U87 cell lines. Interestingly, the effect of 3α-THP was significantly decreased by PP2 (Figure 7 and Appendix A).

## 3. Discussion

In recent years the evidence of sexual dimorphism and gonadal steroids participation in the GB pathophysiology has substantially increased (for review, see [6,35]). Moreover, it has been reported that such hormones are actively metabolized in GB cells [14], and by neural and glial cells in the CNS [19]. This is important because GBs are more frequent in aging patients when the gonadal steroidogenesis is practically null. However, many studies point out that steroidogenic enzymes expression is maintained in different brain areas during aging [36]. Recently, a study based on the transcriptomic information of nearly 900 GB isocitrate dehydrogenase (IDH)-wt patients included in four databases showed that the expression of steroidogenic enzymes by the cells in the core tumor correlates with poor prognostic of GB patients [37].

Despite this, the relevance of many steroid metabolites has been poorly studied in cancer, particularly in glioblastomas. In breast cancer, higher levels of 5α-pregnanes than in normal tissue have been reported, suggesting that progestins like 5α-DHP and 3α-THP have relevance in cancer pathophysiology [38]. In breast cancer cell lines, 5α-pregnanes promote an augment in the cell viability, even in an estrogen receptor (ER)-/PR-cell line, suggesting that such metabolites exert their actions independently of PR [16,39]. Regarding GB, we have reported that P4, 5α-DHP, and 3α-THP promote cell viability and proliferation of human GB cell lines [27,28,40]. Here we report that 3α-THP promotes migration and invasion in three human GB cell lines (Figure 1 and Figure 2). It is important to notice that the concentrations used in this protocol are relatively low and in concordance with physiological concentrations. Our results are similar to those in the human ovarian cancer cell line IGROV-1, where 3α-THP promotes cell migration and invasion [17]. In contrast, other authors have recently reported that high concentrations of 3α-THP (20–60 µM) increased temozolomide-inhibited migration in human T98G and A172 GB cells [41]. This points out that 3α-THP, similar to P4, differently affect GB progression depending on the concentration, at least in vitro [42].

The Aldo-keto reductases AKR1C1-4 modulate the catabolism of many drugs and gonadal steroids. Along with the 5αR, Aldo-keto reductases regulate the levels of many steroids, and therefore, their local availability. In the particular case of P4 metabolism, the reduction to 5α-DHP through the action of 5αR is irreversible. However, the catabolism of 5α-DHP to 3α-THP is reversible. Besides their 3α-HDS activity, AKR1C1-3 isozymes also have 17β- and 20α-HDS activity. We have reported before that GB cell lines express 5αR and AKR1C1-4 at a transcriptional level [18,28]. Since a tiny band was found in the Western blot for either HA or U87 cells, we cannot discard their presence (Figure 3). In a previous work, we have evaluated the expression of such isozymes by RT-qPCR, and we found concordant results between them and those reported in the present study: both HA and U87 cells present the lower expression levels of such enzymes. When compared with other steroidogenic tissues, the expression of all isozymes in the human brain is low, so we expect that the levels of such enzymes, were also low in human astrocytes [32]. It has been reported that the activity of such enzymes is highly cell type dependent in the CNS. In rat-derived cells, greater significant activity of 3α-HSD was detected in a very particular subtype of astrocytes (type 1 astrocytes), rather than in type 2 astrocytes or neurons [43]. We consider these results cannot be entirely extrapolated or compared to humans, because until now, only one AKR1C family member with 3α-HSD activity has been characterized in rats [44]. Besides, many authors determined the activity of such enzymes on different glioma cell lines such as C6 (rat origin), 1231N1 (human astrocytoma), and U87 cells [12,14]. Here we report AKR1C1-4 protein levels in four human GB cell lines compared with those of human astrocyte lysate (Figure 3). Remarkably, the levels of such enzymes were higher in GB cells than in the normal human astrocyte lysate, except for the U87 cell line. Some microarrays analyses suggest that the expression of AKR1C1-3 transcripts levels are significantly lower in GB biopsies compared to normal brain tissue [45]. Discrepancies between AKR1C1-4 mRNA and protein levels have been reported, and they emphasize the importance of better strategies to detect and determine the functional status of such enzymes [46]. The expression of AKR1C1-4 commonly is correlated with a poor prognostic in patients with different types of cancer [34].

As we mentioned, AKR1C1-4 regulates the availability of P4 and its metabolites to activate different signaling pathways. While P4 and 5α-DHP promote the activation of intracellular PR, 3α-THP has no affinity to such receptors [18,27,28]. Previously, the role of PR in migration and invasion has been described [8,40]. Here, the silencing of AKR1C1-3 isozymes was necessary to ensure a low conversion of 3α-THP to metabolites like 5α-DHP with different mechanisms of action from that of 3α-THP. Considering a normal phenotype of AKR1C1-4 enzymes in the GB cell lines, it is expected that isozymes AKR1C1-2 are the main involved in the oxidation of 3α-THP to 5α-DHP. It has been reported that AKR1C3 isozyme has low oxidation activity of 3α-THP [32]. Additionally, 3α-THP is catabolized to 5α-pregnane-3α,20-diol (5α-pregnanediol) by AKR1C1 enzymes [47]. This metabolite possesses activity as a partial agonist of the γ-aminobutyric acid (GABA) receptor type A (GABA_A_R), although with less affinity for the same allosteric site than that of 3α-THP [48]. In this study, we found that AKR1C1-3 silencing did not alter the effect of 3α-THP on GB cell migration and invasion (Figure 4 and Figure 5), which indicates that 3α-THP promotes cell migration and invasion independently of its interconversion to 5α-DHP or other P4 metabolites such as 5α-pregnanediol. Since many mechanisms of action have been detected for 3α-THP, it is important to determine them in glioblastoma. 

The most described mechanisms of 3α-THP so far can be divided into rapid mechanisms mediated by membrane receptors or genomic and slow ones, which depend on the activation of transcription factors. The first implies the modulatory function of neurotransmitters receptors and the activation of membrane P4 receptors (mPR). Regarding rapid mechanisms, 3α-THP is a positive allosteric modulator of the ionotropic receptor type A of GABA neurotransmitter (GABA_A_R); such effect is enhanced in extra-synaptic GABA_A_R containing δ and γ2-3 subunits [49]. Besides, the physiological concentration of 3α-THP regulates other aspects of GABA neurotransmission by promoting the expression of three isoforms of the metabotropic GABA receptor type B (GABA_B_R) in rat Schwann cells from 4 to 24 h of treatment [50]. Also, 3α-THP promotes GABA metabolism by upregulating the expression of two isoforms of the glutamate decarboxylase, the rate-limiting step enzyme of GABA synthesis [51]. It has been suggested that GABA_A_R levels in GB are lower than in low-grade gliomas [52]. 3α-THP also has a high affinity to mPRδ and α subtypes. Such receptors are members of the progestin and adipoQ receptor family. mPRδ is coupled to G_s_ proteins that activate adenylyl cyclase (AC) and favors the increase of AMPc and the activation of ERK in MDA-MB-231 human breast adenocarcinoma cells [53]. With a lesser affinity to that of mPRδ, 3α-THP also binds to mPRα and mPRβ, which are coupled to G_i_ proteins that promote a decrease in AMPc in GT1-7, a mouse hypothalamic GnRH neuronal immortalized cell line [54]. Also, these mechanisms involve the activation of rapid signaling pathways, particularly of c-Src, and PI3K-Akt signaling to promote cell proliferation and migration of human Schwann cell-like differentiated from adipose stem cells [55].

As indicated before, 3α-THP also modulates gene transcription due to the activation of the pregnane xenobiotic receptor (PXR). Such receptors mainly promote the expression of transporters and metabolic enzymes, affecting the biosynthesis of 3α-THP itself [56]. Here, we observed that 3α-THP (100 nM) treatment induced the rapid phosphorylation of c-Src at 10 and 15 min in U251, U87, and LN229 cell lines (Figure 6, Appendix A), although it remains unclear which component of the described 3α-THP mechanisms of action promote the rapid activation of the non-receptor tyrosine kinase c-Src in human GB cell lines. Moreover, when U87 and U251 cell lines were treated with the c-Src inhibitor PP2, the effect of 3α-THP on GB cell invasion was suppressed and dependent on c-Src activation (Figure 7 and Appendix A). This suggests that 3α-THP effect depends on the expression status of its possible effectors. Regarding the normal SNC cells, Melfi and cols. reported that low 3α-THP concentrations promote migration in Schwann rat cells due to activation of c-Src and FAK activation [26].

## 4. Materials and Methods

### 4.1. Cell Culture and Treatments

Human GB cell lines U251, LN229, T98G, and U87 (unknown origin) were purchased from ATCC. U251 and U87 cells were authenticated before by STR profiling. All cell lines were cultured in high glucose phenol red Dulbecco’s Modified Eagle Medium (DMEM, Biowest, FRA) supplemented with 10% Fetal Bovine Serum (FBS, Biowest, FRA), pyruvate 1 mM (InVitro SA, MEX), non-essential amino acids 0.1 mM (InVitro SA, MEX), and 10 mL/L of antibiotic-antimycotic (Catalog number: L0010; Amphotericin B, Penicillin G Sodium Salt, and Streptomycin Sulfate; Biowest, FRA). Cells were maintained in a 5% CO_2_ humidified atmosphere at 37 °C. Before steroid treatments, the medium was replaced by DMEM supplemented with charcoal dextran filtered 10% FBS (without steroid hormones) for 24 h. Cells were treated with vehicle (V; 0.1% ethanol), progesterone (P4; 10 nM), and different concentrations of 3α-THP (10, 100 nM, and 1 µM), or as indicated in each section. P4 was purchased from Sigma Aldrich (St. Louis, MO, USA), and 3α-THP was purchased from MP Biomedicals (Santa Ana, CA, USA). To evaluate the involvement of c-Src in the 3α-THP effects on migration and invasion, c-Src was pharmacologically inhibited with pyrazolopyrimidine (PP2, 1 μM) (Sigma Aldrich, MO, USA).

### 4.2. Migration Assays

To determine the effects of 3α-THP on GB cell migration, wound-healing assays were performed. 3.5 × 10^5^ U251 or U87 cells, and 2.5 × 10^5^ LN229 cells were seeded in 6-well plates. After 24 h of culture in the red-free DMEM supplemented as indicated in the Cell Culture section, the wound was made in the cell monolayer with a 200 µL fine pipette tip. The culture medium and the detached cells were washed out with PBS. The cell culture medium was replaced, and 10 µM of cytosine β-D-arabinofuranoside (AraC) was added 1 h before vehicle and steroid treatments to inhibit cell cycle progression. First, we tested different concentrations of 3α-THP: 10 nM, 100 nM, and 1 µM to determine the concentration with effect on cell migration. For subsequent experiments, the chosen concentration of 3α-THP was 100 nM.

To determine the participation of AKR1C1-4 in 3α-THP on cell migration, the silencing was performed first as indicated in the siRNA Silencing of AKR1C1-3 section. 24 h after the silencing protocol, the migration assays with V, and 3α-THP 100 nM, were carried out.

Immediately after adding treatments, photographs of four different fields were taken with an Infinity 1-2C camera (Lumenera, Ottawa, ON, Canada) coupled to the inverted microscope Olympus CKX41. Then, images of the previously selected fields were taken at 0, 6, 12, and 24 h. Images were processed in the ImageJ software with the MRI Would Healing Tool macro.

### 4.3. Invasion Assays

To determine the effects of 3α-THP on GB cell invasion, we used a modified Boyden chamber assay. Matrigel (1 mg/mL; extracellular matrix of Engelbreth-Holm-Swarn murine sarcoma, Sigma-Aldrich, St. Louis, MO, USA) diluted in phenol red-free DMEM without supplements were added to the membrane inserts (pore: 8.0 µm; Corning, New York, NY, USA) and incubated at 37 °C and 5% CO_2_ atmosphere for 2 h. Phenol red-free DMEM supplemented with 10% FBS was added as a chemoattractant in the lower chamber, and 2 × 10^4^ U251, U87, or LN229 were incubated on the top of the inserts in phenol red-free DMEM without supplements, with cytosine β-D-arabinofuranoside (AraC; 10 µM), and the pharmacological treatments (V, and 3α-THP 100 nM).

To determine the effect of the pharmacological inhibition of c-Src on the 3α-THP effect on cell invasion, cells were treated with V, 3α-THP 100 nM, the c-Src inhibitor PP2 (1 μM), and the conjunct treatments of 3α-THP + PP2.

Silencing of Aldo-keto reductases AKR1C1-3 was performed as indicated in the siRNA Silencing of AKR1C1-3 section before the invasion assays with the pharmacological treatments were performed.

After 24 h of culture at 37 °C and 5% CO_2_ atmosphere, the cell culture medium was retired, and the inserts were gently washed with PBS to eliminate Matrigel and uninvaded cells. Invading cells were fixed to the membrane in the insert with 4% paraformaldehyde for 5 min and stained with crystal violet 1%. After mounting samples with synthetic resin, four fields per treatment were photographed in the Olympus Bx43F microscope (Olympus, Center Valley, PA, USA). Invading cells were counted in the ImageJ software (National Institute of Health, Seattle, WA, USA).

### 4.4. siRNA Silencing of AKR1C1-3

To determine if the effect of 3α-THP on cell migration and invasion is mediated through the Tyrosine-Protein Kinase c-SRC, its expression was silenced with one siRNA. To determine the influence of 3α-THP metabolism on GB cell migration and invasion, AKR1C1-3 were silenced using one siRNA. 2 × 10^5^ U251, U87, or LN229 cells were plated per well in 6-well plates. Cells were grown in phenol red-free DMEM without supplements to perform the transfection. Separately, cells were transfected with a negative control siRNA (control siRNA, 100 nM; Silencer Select Negative Control #1 siRNA1, catalog number: 4390844; Thermo Fisher Scientific, Bannockburn, IL, USA), and the sAKR1C1-3 (100 nM; catalog number: 4392420; ID: s3988; Thermo Fisher Scientific, Bannockburn, IL, USA). The transfection agent was Lipofectamine RNAiMAX (7.5%; Thermo Fisher Scientific, IL, USA); siRNAs were incubated with the transfection agent at room temperature before transfection for 20 min. Then, cells were incubated with siRNA dilutions for 12 h. After transfection, cells were cultured with supplemented phenol red-free DMEM for 24 h. Western blot verification of silencing was performed for the migration or invasion assays.

### 4.5. Protein Extraction and Western Blotting

To determine the protein content of the Aldo-keto reductases AKR1C1-4 in the human GB cell lines, 2 × 10^6^ U251, U87, LN229, and T98G cells were cultured in 100 mm cell culture plates, as mentioned in the cell culture and treatments section. Before the cellular lysis, cells were cultured in red phenol-free DMEM supplemented with 10% charcoal dextran-filtered SFB for 24 h. Cells were collected and homogenized in RIPA buffer (50 mM Tris-HCl pH 7.5, 150 mM NaCl, 1% Triton, 0.01% SDS, and ethylenediaminetetraacetic acid EDTA, 0.5M, 1 mL) with a protease inhibitors mixture (p8340, Sigma-Aldrich, St. Louis, MO, USA). Samples were maintained in agitation at 4 °C for 1 h, and then centrifuged at 14,000 rpm for 15 min. The supernatant was obtained and stored at −20 °C until quantification with the Pierce Protein Assay reagent (Thermo Scientific, Waltham, MA, USA) according to manufacturer’s instructions, and the NanoDrop 2000 spectrophotometer (Thermo Fisher Scientific, IL, USA) at 660 nm. For Western blot determination of AKR1C1-4 (37 kDa), 20 µg of normal human primary astrocytes lysate (HA; 1806, ScienCell, Carlsbad, CA, USA), and 20 µg of protein lysate of the cell lines were mixed with Laemmli 2X buffer (100 mM Tris-base pH 6.8, 20% glycerol, 4% SDS, 10% β-mercaptoethanol, and bromophenol blue) were boiled for 5 min and separated in a 12% SDS-PAGE gels at 80 V. The separated proteins were then transferred to nitrocellulose membranes (Millipore, Burlington, MA, USA) by electrophoresis in semi-dry conditions at 30 mA per membrane for 2 h. Membranes were blocked in agitation at 37 °C with a blocking solution (TBS buffer-0.1% Tween with 5% bovine serum albumin; InVitro, MEX) for 2 h; then, membranes were incubated with a mouse monoclonal AKR1C1-4 antibody (1:1000; sc-390419, Santa Cruz, CA, USA) overnight. Also, blots were incubated with the conjugated to horseradish peroxidase secondary antibody (1:10,000; goat anti-mouse IgG, Santa Cruz, CA, USA) for 45 min. To correct the protein amount loaded in each line, the content of AKR1C1-4 was normalized to that of α-Tubulin. Blots were stripped with a glycine solution (0.1 M, pH 2.5, 0.5% SDS) in agitation for 30 min at 50 °C, and incubated with a mouse anti-α-Tubulin monoclonal primary antibody (T9026, Sigma Aldrich, St. Louis, MO, USA) overnight, and with the goat anti-mouse secondary antibody in agitation for 45 min. For AKR1C1-4 silencing, the same Western Blot conditions were used. 

To determine c-Src in GB cell lines, 30 μg of total protein were separated in 8.5% SDS-PAGE gels at 80 V, transference to a nitrocellulose membrane was performed at semi-dry conditions at 25 V for 30 min. When c-Src activation was evaluated, the membranes were first incubated with the primary antibodies for the phosphorylated or total forms of c-Src: phospho c-Src Tyr-416 (1:1000; Ref. 2101; Cell Signaling, Massachusetts, MA, USA), and c-Src (1:1000; Ref. 2108, Cell Signaling, Massachusetts, MA, USA). In both cases, blots were incubated with the conjugated to horseradish peroxidase secondary antibody (1:10,000; Ref: 1858415, goat anti-rabbit IgG, Thermo Scientific, Waltham, MA, USA). The total content of proteins was normalized to that of α-Tubulin.

Chemiluminescence signals of the membranes were detected with the SuperSignal West Femto Maximum Sensitivity (Thermo Scientific, Waltham, MA, USA) kit, according to the manufacturer’s instructions, and in Kodak Biomax Light Films (Sigma-Aldrich, St. Louis, MO, USA). Western blot images were captured with a digital camera (SD1400IS, Canon), and densitometric analysis was performed in the ImageJ software (National Institute of Health, Seattle, WA, USA).

### 4.6. Statistical Analysis

All graphs and statistical analysis were performed in GraphPad Prism 8 Software (GraphPad Software Inc., La Jolla, CA, USA). Each experiment was performed at least in triplicate or as indicated in each figure legend. A one-way ANOVA and the post hoc Bonferroni test with a confidence interval of 95% were performed. A value of *p* < 0.05 was considered significant and was indicated in each figure.

## 5. Conclusions

Our work indicates that 3α-THP at low concentration participates in the regulation of glioblastoma (GB) malignancy by promoting cell migration and invasion. Such effects are mediated by the rapid activation of c-Src. Besides, GB cells express AKR1C1-4 isozymes, involved in the metabolism of 3α-THP and other steroids, which indicate that such metabolites are actively generated by GB. These data point out the relevance of P4 metabolites in the pathophysiology of GB since they promote the rapid progression of GB through different mechanisms from that of P4.

## Figures and Tables

**Figure 1 ijms-23-04996-f001:**
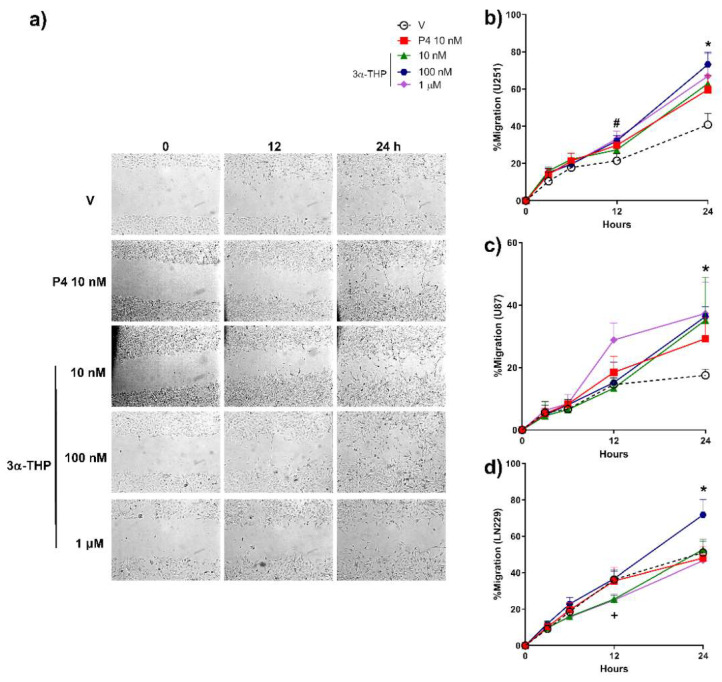
Effect of 3α-THP on human GB cell migration. (**a**) Representative images of the scratch area from U251 cells treated with vehicle (V, ethanol 0.1% in the medium), progesterone (P4; 10 nM), or allopregnanolone (3α-THP; 10, 100 nM and 1 µM). Percentage migration graphs of (**b**) U251, (**c**) U87, and (**d**) LN229 human GB cell lines. Each point represents the mean ± SEM. U251 (Graph b): # *p* < 0.05 for P4 and 3α-THP 100 nM vs. V; * *p* < 0.05 for P4 and all concentrations of 3α-THP vs. V. U87 (Graph c): * *p* < 0.05 for 3α-THP 10 nM, 100 nM and 1 µM vs. V. LN229 (Graph d): + *p* < 0.05 3α-THP 10 nM and 1 µM vs. V; * *p* < 0.05 for 3α-THP 100 nM vs. all other treatments. *n* = 3 for U251 and U87 cells; *n* = 4 for LN229 cell line.

**Figure 2 ijms-23-04996-f002:**
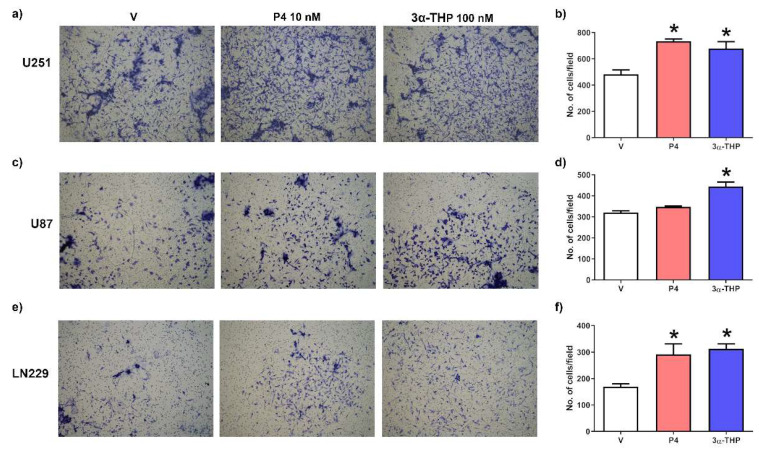
Effect of 3α-THP on the invasion of human GB cell lines. Cells were treated with vehicle (V; ethanol, 0.1% in the medium), P4 (10 nM), and 3α-THP (100 nM) for 24 h. Representative images of (**a**) U251; (**c**) U87, and (**e**) LN229 invasion assays. All photographs were taken at 10× augment. Graphs of the invading cells number per field: (**b**) U251; (**d**) U87; and (**f**) LN229 cell lines. Each column represents the mean ± SEM. *n* = 3 for all cell lines evaluated; * *p* < 0.05 vs. V.

**Figure 3 ijms-23-04996-f003:**
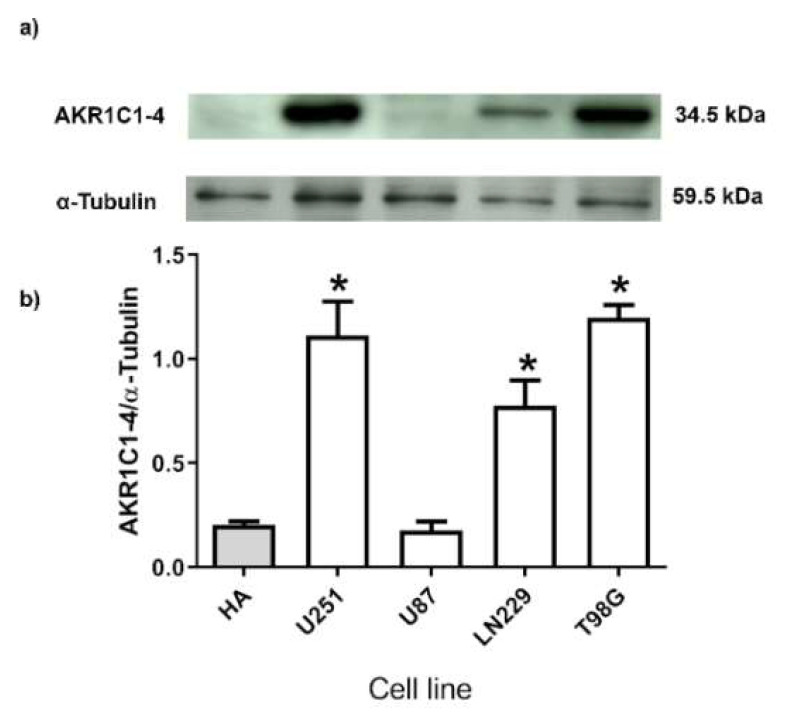
AKR1C1-4 are differentially expressed in GB cells. The expression of AKR1C1-4 was detected in a total protein extract from normal human astrocytes (HA) and different human GB cell lines. (**a**) Representative Western blots of AKR1C1-4 and α-Tubulin, which was used as a loading control. (**b**) Densitometric analysis graph. Each column represents the mean ± SEM. *n* = 3; * *p* < 0.05 U251, T98G, and LN229 vs. HA and U87 cell lines.

**Figure 4 ijms-23-04996-f004:**
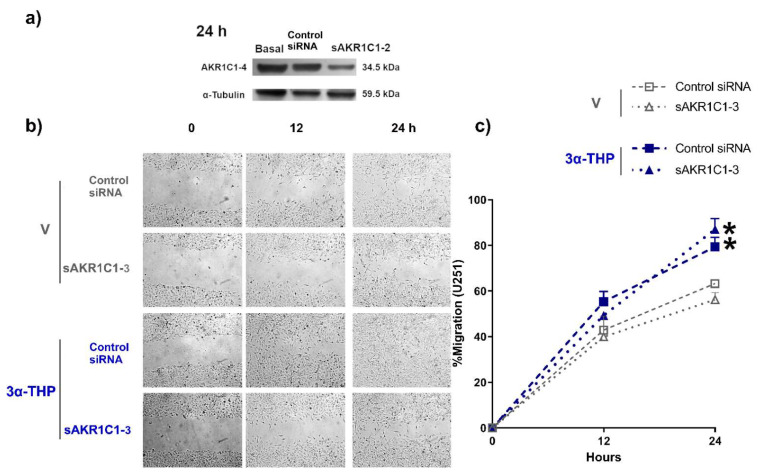
Effect of 3α-THP and the silencing of AKR1Cs on the migration of U251 cells. (**a**) AKR1C1-3 silencing for 24 h and at the beginning of migration assays. (**b**) Representative images of U251 cells without transfection, transfected with the negative control siRNA (Control siRNA), and transfected with de the siRNA against AKR1C1-3 (sAKR1C1-3). Cells were treated with V or 3α-THP. (**c**) Graph of the percentage of U251 migrating cells. Each point represents the mean ± SEM. * *p* < 0.05 3α-THP (control siRNA, and sAKR1C1-3) vs. V (control siRNA, and sAKR1C1-3); *n* = 3.

**Figure 5 ijms-23-04996-f005:**
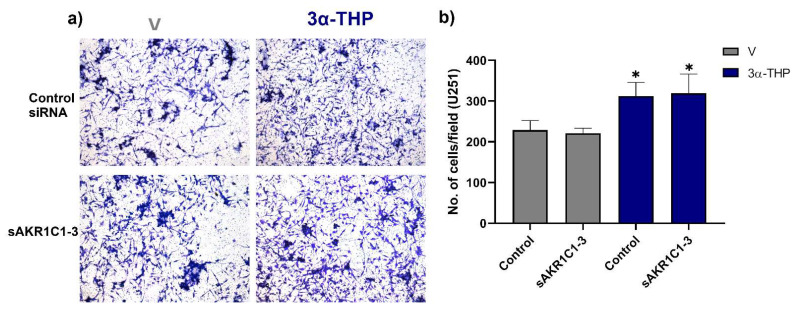
Effect of 3α-THP and the silencing of AKR1Cs on the invasion of U251 cells. (**a**) Representative images of U251 cells transfected with the negative control siRNA (Control siRNA), the siRNA against AKR1C1-3 (sAKR1C1-3) and treated with V or 3α-THP. (**b**) Graph of the number of invading U251 cells per field. Each point represents the mean ± SEM. * *p* < 0.05 THP (control SiRNA and sAKR1C1-3) vs. V (control siRNA and sAKR1C1-3) *n* = 4.

**Figure 6 ijms-23-04996-f006:**
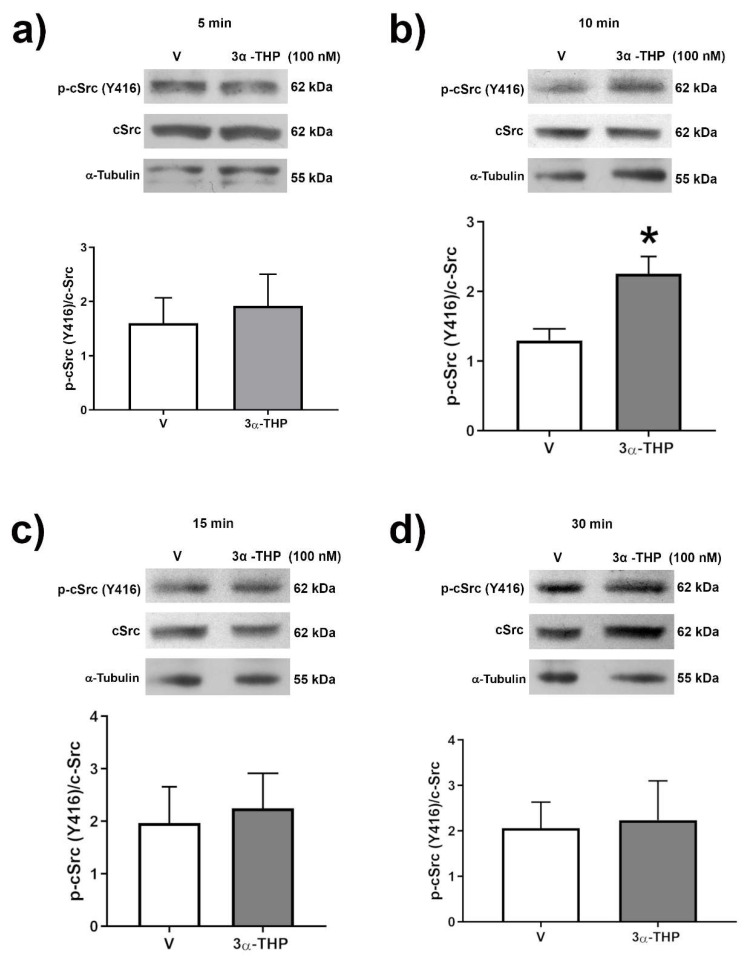
Effect of 3α-THP on c-Src activation in U251 GB cells in time-course experiments. The phosphorylation of c-Src (Y416) was determined under the treatments of 3α-THP 100 nM or Vehicle (V, EtOH 0.01%) at (**a**) 5 min; (**b**) 10 min; (**c**) 15 min; (**d**) 30 min by Western blot. The upper panel of each section shows a representative Western blot experiment of p-c-Src, total c-Src, and α-Tubulin as a loading control. Each lower panel presents the densitometric analysis of p-c-Src/c-Src. Each column represents the mean ± SEM., *n* = 3; * *p* < 0.05 3α-THP vs. V.

**Figure 7 ijms-23-04996-f007:**
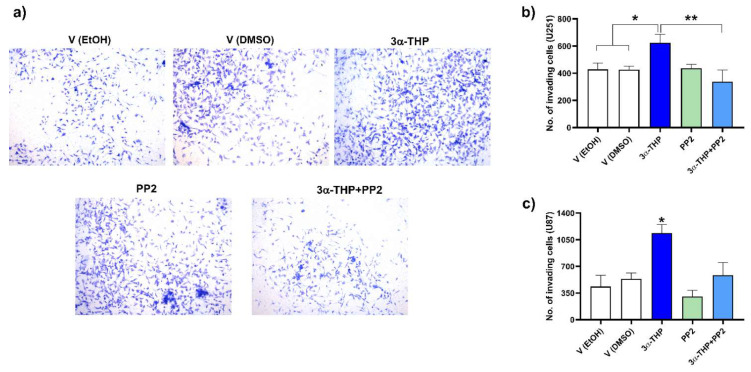
Effect of 3α-THP and the pharmacological inhibitor of c-SRC PP2 on U251 and U87 cell invasion. (**a**) Representative images of U251 cell invasion assays. All photographs were taken at 10× augment. (**b**) Graph of the number of invading U251 cells per field. Each column represents the mean ± SEM., *n* = 4; * *p* < 0.05 3α-THP vs. V (EtOH, and DMSO); ** *p* < 0.005 3α-THP vs. 3α-THP + PP2. (**c**) Graph of the number of invading U87 cells per field. Each column represents the mean ± SEM., *n* = 4; * *p* < 0.05 3α-THP vs. all other treatments.

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
