# Peer review of "Allopregnanolone Promotes Migration and Invasion of Human Glioblastoma Cells through the Protein Tyrosine Kinase c-Src Activation"

_ijms, 2022, doi:10.3390/ijms23094996_

Round 1

Reviewer 1 Report

The authors try to discuss in the manuscript effect of allopregnanolone on migration and invasion of human glioblastoma.  The manuscript is interesting, but minor revision is important:

INTRODUCTION

Line 29 should be “in the central”

Line 30 please delete “of”

Line 54 should be “the cellular”

Line 68 should be “is poorly”

Line 31-33 it should be added from which year are presented data. The same – line 35.

RESULTS

Line 135 should be “the invasion”

Fig.2 I suggest changing that symbol below the “star” showing statistical because in present form it suggests that results P4 10nm were compared with 3a-THP 100nm as well as the differences are statistically significant. While the results should be compare with control. Moreover, legend below the graphs should be more readable.

Fig.3 how the authors will explain that normal astrocytes and U87 did not express AKR1C1-4? Is it possible that all 4 analyzed proteins have strictly the same molecular mass? I ask because even if mass is similar 32-46 kDa, the high of bands on WB membrane should be greater (similar to the photo presented by the manufactererer).

Fig 6a what is a reason of occurring 2 bands on a-tubuline? Please explain.

DISCUSSION

Line 209 should be “have to”

Line 221 should be “regulates”

Line 225 should be “were”

Line 228 “productors” is it ok?

Line 266 should be “promote”

Line 269 should be “suggests”

Line 271 should be “promote”

MATERIALS and METHODS

Line 300,310 should be “were”

Line 313 should be “on the top”

Point 4.1 Regarding U87 MG cell line used in your study, please clarify whether you refer to the original glioblastoma cell line established in the University of Uppsala (https://web.expasy.org/cellosaurus/CVCL_GP63) or the U87 MG ATCC version and which is most probably a glioblastoma but whose origin is unknown (https://web.expasy.org/cellosaurus/CVCL_0022).
In case it is the ATCC version, please rephrase to "glioblastoma of unknown origin" in the Materials and methods section of your manuscript and provide the catalogue number of the cell line. If you have authenticated the cell line used in your study, please add a statement of authentication, including which method was used for authentication (e.g. STR profiling or other method) in the Materials and methods section of your manuscript. This statement is optional, however, we strongly encourage you to add it.

Moreover, if the authors use any antibiotics or antifungal compounds in the used medium it must be mentioned (used concentration and manufacturer).

Reviewer 2 Report

Allopregnanolone promotes migration and invasion of human glioblastoma cells through c-Src activation

By Zamora Sánchez et al.

General comment

This is a new work of the research group based on the behavior of estrogens in the development of glioblastoma, a severe type of brain cancer. On this occasion, the authors describe the invasive role of a progesterone derivative, allopregnanolone, relating this migration to the activation of tyrosine kinase c.Src after administration of this neurosteroid in several glioblastoma cell lines.

Basically, there are two types of work in research, one, the authors seek effective solutions to a problem with their work, and the other, the authors seek reasons why a certain problem develops, and this is of this second type. The interest of this second type of scientific approach is that once the problem is known, it will be a little easier to try to solve it.

After a critical and exhaustive reading of the manuscript and other related ones, I have to say that this is a serious, analytical work, with an excellent methodological approach for what is sought and with results in accordance with the hypothetical approach at the beginning.

Assuming that what is proposed by the authors is not the only molecular mechanism that explains the invasion and migration of glioblastoma, it is certain that they do reveal at least one of the responsible mechanisms, such as the role of c-Src kinase (Proto-oncogene tyrosine-protein kinase) in the development of this pathology.

Background comments

The authors base many of their conclusions on the significant differences between control values and the problems caused by allopregnanolone administration, and although this is true in some instances, it does not appear to be in the majority of them:

  1. Figure 2, sections b and f. It is difficult to believe that there are significant differences between P4 and allopregnanolone, considering the level of the errors and the number of observations. Therefore, it would be advisable for the authors to revise the values and in any case to put the individual results in the supplementary material section.
  2. Figure 4, section c. Same comment.
  3. Figure 5, section b. Same comment.
  4. In any case, I believe that the authors have not succeeded in putting in the graphs, between which groups there are significant differences.

Therefore, the authors should give a clear explanation on this point.

Comments on form

1.- At present, the use of abbreviations is too often abused, and this abuse makes reading comprehension very difficult for readers of the work who are not very familiar with these abbreviations, and although it is true that, on most occasions, but not all, the authors clarify the meaning at the beginning,

  1. it would be convenient from time to time, since it is easy to do so, to put the meaning instead of the abbreviation.
  2. At the end of the article, put a list of the abbreviations used in the text.

2.- Title: The authors should clarify in the title that c-Src is a tyrosine kinase.

3.- Figure captions should not be centered.

4.- Although the work is focused on in vitro experiments, it would be very interesting to observe if what happens in vitro also happens in vivo.
